# Targeting Exosomal PD-L1 as a New Frontier in Cancer Immunotherapy

**DOI:** 10.3390/cimb47070525

**Published:** 2025-07-08

**Authors:** Laura Denisa Dragu, Mihaela Chivu-Economescu, Ioana Madalina Pitica, Lilia Matei, Coralia Bleotu, Carmen Cristina Diaconu, Laura Georgiana Necula

**Affiliations:** 1Stefan S. Nicolau Institute of Virology, 030304 Bucharest, Romania; denisa.dragu@virology.ro (L.D.D.); ioana.pitica@virology.ro (I.M.P.); lilia.matei@virology.ro (L.M.); coralia.bleotu@virology.ro (C.B.); carmen.diaconu@virology.ro (C.C.D.); laura.necula@virology.ro (L.G.N.); 2Faculty of Medicine, Titu Maiorescu University, 040051 Bucharest, Romania

**Keywords:** ExoPD-L1, exosome, immune evasion, cancer immunotherapy, therapy resistance

## Abstract

This manuscript assesses the critical role of exosomal PD-L1 (ExoPD-L1) in immune suppression, tumor progression, and resistance to therapy. ExoPD-L1 has been identified as a key mediator of tumor immune evasion, contributing to systemic immunosuppression beyond the tumor microenvironment (TME) due to its capacity to travel to distant anatomical sites. In this context, the review aims to elaborate on the mechanisms by which exosomal PD-L1 interacts with T cell receptors and modulates both the tumor microenvironment and immune responses, impacting patient outcomes. We further explore emerging therapeutic strategies that target ExoPD-L1 to enhance the effectiveness of immunotherapy. Blocking ExoPD-L1 offers a novel approach to counteracting immune escape in cancer. Promising strategies include inhibiting exosome biogenesis with GW4869 or Rab inhibitors, neutralizing ExoPD-L1 with targeted antibodies, and silencing PD-L1 expression through RNA interference (RNAi) or CRISPR-based methods. While each approach presents certain limitations, their integration into combination therapies holds significant potential to improve the efficacy of immune checkpoint inhibitors. Future research should focus on optimizing these strategies for clinical application, with particular attention to improving delivery specificity and minimizing off-target effects.

## 1. Introduction

Exosomes are membranous nanovesicles secreted into the extracellular space by almost all cell types that can deliver various cellular components to adjacent or distant cells, participating in the regulation of various pathophysiological processes. Lately, it has been recognized that exosomes play an essential role in the process of carcinogenesis, tumor progression, angiogenesis, metastasis, and therapy response [1,2,3]. Accumulating evidence has shown that PD-L1 is highly expressed on the surface of tumor cell-derived exosomes (TEX) and that exosomal PD-L1 (ExoPD-L1) shares the same membrane topology as cell surface PD-L1, with its extracellular domain displayed on the surface of the exosomes, which can directly bind the PD-1 receptor, inhibiting anti-tumor immune responses [4,5]. Interestingly, the PD-L1 expression levels on TEX are heterogeneous, showing significant variability among different cell lines or types of tumors. Usually, the ExoPD-L1 level correlates with PD-L1 levels in parental cells [4]. However, high amounts of ExoPD-L1 can be secreted by PD-L1-low or absent parental cells. These variations may be related to different packaging capacities of the cells or their metastatic capacity, with malignant cells releasing higher quantities of ExoPD-L1 [6,7]. Additionally, treatment with interferon-gamma (IFN-γ) increases both cell-surface membrane PD-L1 and ExoPD-L1 without affecting the number of vesicles secreted [6,8].

In the last decade, numerous studies have been dedicated to elucidate the role of exosomes in tumor progression, emphasizing their importance in communication between tumor cells and components of the tumor microenvironment (TME), particularly immune system cells. ExoPD-L1 facilitates the communication between tumor cells and immune cells, participating in TME reprogramming and modulating immune surveillance [9,10]. Within the tumor microenvironment, besides tumor cells, different immune cells also express PD-L1. Moreover, it was shown that PD-L1 can be transferred from PD-L1-positive breast cancer cells to PD-L1-negative ones and also to immune cells such as macrophages and dendritic cells (DCs). PD-L1 transported by exosomes was located on the surface of target cells and was able to bind to PD-1. These results indicated that exosomes are capable of transferring functional PD-L1 to other cells [11]. This transfer highlights the function of exosomes as carriers that transport PD-L1 to various immune cells, altering their function and affecting immune surveillance while promoting tumor growth and metastasis. In cancer patients, elevated levels of circulating ExoPD-L1 were correlated with pathological features of the patients [4,11,12,13], and contribute to systemic immunosuppression and resistance to anti–PD-L1 therapy [14].

Recent studies suggest that high levels of ExoPD-L1 secreted by tumor cells are often associated with therapy resistance. ExoPD-L1 can bind therapeutic antibodies and block T cell reactivation that sustain immune evasion, poor treatment response, and worse outcomes, reducing the efficacy of anti-PD-1/PD-L1 therapies [6]. As the exoPD-L1 evaluation might predict the treatment results, the investigation of exosome-targeted strategies has become essential to improve immunotherapy outcomes. Moreover, targeting exosomal PD-L1 could work synergistically with PD-L1 antibodies, highlighting exosomal PD-L1 as a promising therapeutic target to overcome resistance to current immune checkpoint blockade.

## 2. Biogenesis and Molecular Composition of Exosomes

### 2.1. Exosome Formation: Mechanisms of Exosome Biogenesis

Exosomes are lipid bilayer vesicles, about 30–150 nm in diameter, released from both normal and tumor cells through exocytosis. They are found in various bodily fluids like blood, cerebrospinal fluid, urine, saliva, and breast milk. Based on the originating cell, their contents may include nucleic acids, proteins, lipids, amino acids, cell-surface proteins, and other biologically active substances, and may influence different biological functions, such as intercellular communication, antigen presentation, immune response, and extracellular environment regulation [15,16,17]. Due to their abundance and presence in circulation, exosomes are being explored as potential cancer biomarkers for diagnostic and prognostic purposes, as therapeutic targets, or even as carriers for anticancer drugs.

Exosome formation occurs through the endocytic pathway and begins with plasma membrane invagination and the formation of early secretory endosomes.

Specifically for ExoPD-L1, data suggest that it originates from the plasma membrane of the donor cells instead of being derived from the endoplasmic reticulum (ER) or Golgi apparatus [6], and the levels of ExoPD-L1 usually correlate with PD-L1 expression on the membrane of donor cells [18,19].

It has recently been shown that membrane-bound PD-L1 undergoes a constitutive process of endosomal trafficking. After internalization of the plasma membrane and formation of early endosomes, PD-L1 can be transferred to the recycling endosomes or to multivesicular bodies (MVB)/late endosomes and, from here, can be directed either to lysosomes or can be released through EVs [20]. Key enzymes involved in the molecular mechanisms of ExoPD-L1 production include ESCRT, Rab27a, nSMase2, and ALIX, as determined by loss-of-function experiments [6]. Further, it has been shown that PD-L1 endosomal trafficking was regulated by Rab5 and that Rab27 functions in MVB and is involved in exosomal secretion [20].

### 2.2. Exosome Release and Uptake

Exosome uptake by recipient cells is a complex process involving surface interactions, membrane fusion, and various internalization pathways such as endocytosis and phagocytosis. Although the exact mechanism has not been fully elucidated, there is evidence that exosomal surface features—particularly their lipid and protein composition—play a critical role in determining cellular targeting [21,22]. Specific adhesion molecules and integrin profiles on exosomes influence their affinity for certain cell types and tissues. For example, integrins α6β4 and α6β1 are linked to lung targeting via interactions with fibroblasts and epithelial cells, while αvβ5 promotes liver targeting by binding to Kupffer cells [23].

ExoPD-L1 expression varies significantly between different tumor types and also among different cell lines of the same tumor type. Typically, the levels of ExoPD-L1 reflect those present in the corresponding parental tumor cells. Nonetheless, certain exceptions have been noted, including prostate cancer cells, where the tumor cells produce high amounts of exosomes containing PD-L1 but lack cell-surface PD-L1 despite maintaining consistently high levels of PD-L1 mRNA [6]. Also, the metastatic capacity of tumor cells can influence the amount of ExoPD-L1 released in circulation. A recent study reported that metastatic melanoma discharges higher levels of ExoPD-L1 compared to primary melanoma cells [4]. Similarly, the prostate cancer cell lines were found to secrete increased level of ExoPD-L1 in more malignant cells [24].

External factors can also influence the biogenesis of exosomes, including cell type, exposure to certain cytokines, growth factors, hypoxia, calcium signaling, drugs, etc. IFN-γ can increase the production of ExoPD-L1 by activating the JAK/STAT signaling pathway and IRF-1 [6,25,26], and transforming growth factor-beta (TGF-β) was shown to increase the expression level of PD-L1 on the exosomes released by breast cancer cells in a dose-dependent manner [27]. Calcium signaling influences exosome biogenesis and secretion by modulating the Rab GTPase family and membrane fusion factors [28], while hypoxia promotes the release of exosomes by breast cancer cells, and nasopharyngeal carcinoma cells through the hypoxia-inducible factor (HIF-1α) and a complex communication between cancer and immune cells [29,30]. Also, drugs, such as 5-fluorouracil (5-FU), induced upregulation of ExoPD-L1 in gastric cancer cells, probably via the miR-940/Cbl-b/STAT5A axis [31].

## 3. Mechanisms of Exosomal PD-L1 in Immune Suppression

Exosomes are important mediators between tumor cells and immune cells, which can have both negative and positive implications for tumor immunity, promoting or suppressing tumor progression. The precise role that ExoPD-L1 plays in the process of immune evasion in cancer is not fully understood, but various mechanisms, including suppression of T cell activity, modulation of other immune cell populations such as macrophages, DCs, natural killer (NK) cells, and neutrophils, as well as supporting an immunosuppressive TME, are being studied intensively (Figure 1).

### 3.1. Inhibition of T Cell Proliferation and Cytokine Production

Studies indicate that ExoPD-L1 binds to T cells, with PD-L1 playing a complex role in regulating T cell responses [4,11]. Upon engagement with PD-1 receptor on T cells, PD-L1 triggers SHP2-mediated dephosphorylation of the T cell receptor (TCR) and CD28, suppressing antigen-induced T cell activation and contributing to immune evasion [32,33].

By binding to PD-1, PD-L1 induces conformational changes in PD-1 that decrease T cell activation and promotes a distinct cytokine secretion profile characterized by lower expression of IL-2, TNF-α, IFN-γ, and granzyme B (GZMB) [34].

Also, inhibitory signals such as PD-1, CTLA-4, TIM-3, and LAG-3 are released, limiting T cell effector functions in peripheral tissues and tumor microenvironments and contributing to the maintenance of T cell anergy and exhaustion [35,36].

In vitro studies demonstrated that PD-L1 found on tumor-derived extracellular vesicles can activate cAMP-response element binding protein (CREB) and signal transducer and activator of transcription (STAT) signaling that induce senescence and suppression in T cells by a mechanism that mainly targets the DNA damage and hyperactivation of the lipid metabolism that increase the cholesterol level and lipid droplet formation [37].

In nasopharyngeal cancer, it was shown that tumor-derived ExoPD-L1 promotes immune escape by inhibiting the activity of CD8+ T cells through the interaction of ExoPD-L1 with the PD-1 on the surface of T cells [38]. In another study, the non-small cell lung cancer (NSCLC)-derived ExoPD-L1 induced apoptosis and inhibited cytokine production in CD8+ T cells [8].

Overexpression of exoPD-L1 is correlated with advanced tumor staging and sustains tumor growth and metastasis. Therefore, introducing tumor-derived exosomes exogenously enhances both metastatic capacity and the overall burden of primary tumors [39].

By in vitro and in vivo studies, a high level of ExoPD-L1 was identified in metastatic melanoma cells being upregulated and positively correlated with IFN-γ that is secreted by activated T cells and this increased expression can suppress the function of CD8 T cells and facilitates tumor growth. Moreover, ExoPD-L1 expression was significantly higher in metastatic melanoma patients compared to healthy donors, even if there is no difference between the number of circulating exosomes in these two groups. Interestingly, exoPD-L1 expression is higher in metastatic melanoma cells compared to primary melanoma cells [4].

In prostate cancer, the cells with high ExoPD-L1 expression can influence the behavior of prostate cancer cells with low PD-L1 expression, remodulating TME, and protecting them against T cell killing and promoting tumor progression [24].

At the level of TME, ExoPD-L1 exhibits immunosuppressive capacities not only by suppression of T cell activity but also by sustaining the proliferation of immunosuppressive cells [40]. Moreover, TEX delivers negative signals to immune effector cells interfering with their anti-tumor functions, suppressing their functions, leading to tumor progression, and facilitating tumor escape from the immune system [41,42].

In gastric cancer cells, ExoPD-L1 overexpression can stimulate myeloid-derived suppressor cell proliferation by triggering the IL-6/STAT3 signaling pathway [40].

TME immunosuppression can also be supported by the expansion of regulatory T cells (Treg), precursors of follicular regulatory T cells (Tfr). In esophageal cancer, Exo-PDL1 release from cancer cells causes an imbalance in the Tfh/Tfr (follicular helper T cells) cell ratio, promoting the expansion and suppressive functions of Tfr cells and leading to the impairment of anti-tumor T cell functions and an immunosuppressive environment [43].

### 3.2. Suppression of Dendritic Cell Maturation and Antigen Presentation

Subsequently, PD-1 was identified as a specific surface receptor on DCs [44], where it was shown to promote both the induction and maintenance of T cell anergy. Furthermore, in a recent study, it was shown that TEX inhibited the maturation and migration of DCs and the secretion of pro-inflammatory factors, thereby enhancing their immunosuppressive functions. Specifically, treatment with tumor exosomes significantly reduced the differentiation into CD4+IFN-γ+ Th1 cells and increased Treg differentiation. Also, the immunosuppressive capacity of DCs exposed to tumor exosomes was partially reversed upon blockade of PD-L1, indicating that PD-L1 plays a key role in mediating tumor exosome-induced immune suppression by affecting DC function [45].

### 3.3. Polarization of Macrophages Toward the M2 Phenotype and Downregulation of NK Cell Cytotoxicity

In addition to its effects on DCs and T cells, PD-L1 also modulates macrophage function. The impact of tumor-derived exosomes in modulating macrophage polarization was pointed out in recent studies [46,47]. The data showed that the exosomes secreted by lung tumor cells can be internalized by M0 macrophages, inducing transcriptional changes, reprogramming the macrophage metabolism, and stimulating their differentiation into the M2 phenotype, promoting tumor growth and immune suppression [46]. In addition, macrophage M2 polarization induced by ExoPD-L1 was shown to promote the viability, migration, invasion, and epithelial–mesenchymal transition process in lung cancer cells [47]. In addition, exosomes derived from colorectal cancer stimulate macrophage proliferation and elevate their PD-L1 levels, affecting CD4+ T cell function and accelerating cancer progression [10]. Data suggest that PD-L1 expression on macrophages plays a key regulatory role in controlling IFN–γ–mediated NO production by naïve CD4+ T cells, further highlighting the immunomodulatory potential of the PD-1/PD-L1 axis across multiple immune cell types. Elevated levels of circulating ExoPD-L1 are linked to systemic immunosuppression and predict poor response to anti–PD–1 therapy in melanoma patients. ExoPD-L1 from sulfasalazine (SAS)-treated melanoma cells promotes M2 macrophage polarization by upregulating PD-L1 expression, contributing to immune evasion and resistance to PD-1/PD-L1 blockade [48]. Another study revealed the implication of ExoPD-L1 in acute myeloid leukemia (AML) immune evasion by suppressing natural killer (NK) cell activation and inhibiting their cytotoxicity, probably through the activation of the PD-1/PD-L1 pathway [49]. Recently, the role of hypoxia on the immunosuppressive phenotype of macrophages and subsequently on the effector functions of CD8+ T cells was demonstrated [30].

## 4. Exosomal PD-L1 Contribution to Therapy Resistance

In the last few years, numerous studies have suggested that the use of immune checkpoint therapies as first-line treatment can represent an important therapeutic approach in the case of many aggressive cancers, such as advanced melanoma, metastatic NSCLC, and Hodgkin lymphoma.

Nevertheless, there are several cases in which the clinical response to therapy is low, and the exosomal expression of PD-L1 seems to be closely related to the failure of immune checkpoint therapy [39]. The exosomes resulting from cancer cells can control or modify the tumor microenvironment by releasing molecules, such as microRNAs, mRNAs, and proteins that promote tumor progression, metastasis, and resistance to therapy [9]. After the expression of PD-L1 in exosomes was demonstrated, several studies have focused on PD-L1-mediated immune evasion and the correlation between ExoPD-L1 expression and treatment response. The clinical relevance of plasma circulating ExoPD-L1 has been observed in patients with head and neck cancer [50], gastric cancer [51], NSCLC [8], pancreatic cancer [52], and melanoma [4]. In melanoma patients, higher pretreatment levels of circulating ExoPD-L1 were negatively correlated with disease responses to immunotherapy, and, moreover, were linked to poorer clinical outcomes, supporting the critical impact of ExoPD-L1 in therapeutic resistance to anti-PD-L1 antibody treatment. It was suggested that elevated levels of ExoPD-L1 indicate complete exhaustion of T cells making them unresponsive to immunotherapy [6]. Therefore, the evaluation of circulating ExoPD-L1 might be used to predict the efficacy of immunotherapy. In metastatic melanoma patients, evaluation of circulating ExoPD-L1 before and on pembrolizumab treatment suggested a negative correlation between ExoPD-L1 and the activation of anti-tumor immunity. Therefore, an increased level of ExoPD-L1 can block the reactivation of T cells that normally occurs after the anti-PD-1 treatment [4]. Another study showed that the expression of PD-L1 in the plasma of melanoma patients is higher in exosomes compared to soluble PD-L1, and although the ExoPD-L1 is found in all patients, only 67% of tumor biopsies are PD-L1 positive. Therefore, tumor cells prefer to secrete PD-L1 using the exosomes, while a lower expression is found on the cell surface [6,53]. In head and neck cancer patients treated with cetuximab (EGFR inhibitor), ipilimumab (immune checkpoint inhibitor), and radiation therapy, the levels of CD3(+)PD-L1+ exosomes are significantly higher in patients whose tumors recurred, as compared to those who remained disease-free [54]. These studies suggest that an increased level of ExoPD-L1, either at baseline or during treatment, can have a negative impact on anti-PD-L1 therapy, but the patients with the highest levels of circulating pre-treatment PD-L1+ exosomes appear to be the most affected [4].

Metastatic NSCLC PD-L1 ≥50% patients treated with combination therapy, that includes ICI and chemotherapy, had a higher median overall survival compared with monotherapy and reduced the risk of overall mortality. However, a better refinement of the patient selection criteria may improve the therapeutic outcomes and enhance the overall survival [55,56].

These discoveries highlight the involvement of ExoPD-L1 in mediating resistance to current anti-PD-L1/PD-1 therapies, but the mechanisms that sustain this resistance remain unclear. One possible mechanism is the reduction in the amount of antibodies that successfully target PD-L1 expressed on tumor cell surfaces by binding to ExoPD-L1 [57]. Taking into account these observations, it becomes essential to identify the factors that mediate the higher or lower ExoPD-L1 levels at baseline or during treatment in different cancer patients, and more importantly, to evaluate exosome depletion as adjuvant therapy that sustains the effects of anti-PD-1 therapy [54].

Another aspect that needs to be taken into account is the high level of drug resistance and the severe side effects associated with the anti-PD-L1 monoclonal antibody, which might be overcome by developing new peptide-based and non-peptide small molecule PD-L1 inhibitors as an alternative strategy to improve therapy outcome [58].

## 5. Therapeutic Strategies Targeting Exosomal PD-L1

ExoPD-L1 has been identified as a key mediator of tumor immune evasion, participating in systemic immunosuppression beyond the TME, due to its ability to travel to distant sites. An increasing number of preclinical and clinical studies highlight the significant role of ExoPD-L1 in the relatively low response rates to anti-PD-L1/PD-1 therapy in certain tumors. Consequently, the idea that the removal of ExoPD-L1 can increase the response rate to the anti-PD-L1 blockade has emerged and has been further supported by preclinical studies. Current strategies to block ExoPD-L1 activity include inhibiting exosome secretion, neutralizing ExoPD-L1 directly, and silencing PD-L1 expression through genetic or RNA-based approaches (Figure 2).

### 5.1. Targeting Exosome Biogenesis and Secretion

One of the most effective ways to block ExoPD-L1 is by using pharmacological agents that reduce the level of exosomes by inhibiting different molecules involved in the generation, packaging, or release of exosomes [59]. However, this aspect is challenging, as exosomes are important for normal cellular functioning and cell-to-cell communication. Pharmacological inhibitors, such as GW4869, which targets neutral sphingomyelinase 2 (nSMase2)—the enzyme that generates the bioactive lipid ceramide involved in cellular processes such as exosome generation [60]—have shown significant promise. GW4869 disrupts the ceramide-dependent ILV formation within multivesicular endosomes (MVEs), thereby reducing exosome release. Preclinical studies have demonstrated that GW4869 reduces ExoPD-L1 secretion, restores T cell activity, and enhances anti-tumor immunity in models of melanoma [61]. Poggio M. et al. reported similar findings in a mouse model of prostate cancer resistant to anti-PD-L1 therapy, using GW4869 treatment to suppress tumor growth by blocking the secretion of immunosuppressive exosomes [6]. In addition to GW4869, Rab GTPase inhibitors target key regulators of exosome trafficking, being involved in the fusion of the MVE to the plasma membrane. Several Rab GTPase inhibitors are known, such as Nexinhib20, Tipifarnib, and Y-27632, which are small-molecule agents that functionally converge as Rab27A pathway inhibitors—either by directly disrupting Rab27A–effector interactions (as in the case of Nexinhib20), targeting the Rab27A WF pocket through structure-based modeling (Tipifarnib), or indirectly modulating exosome release through upstream signaling pathways like ROCK inhibition (Y-27632) (Table 1). The “WF pocket” of Rab27A refers to a specific binding site on the surface of the Rab27A GTPase that is critical for interaction with proteins regulating exosome secretion. The term WF derives from two conserved hydrophobic residues, tryptophan (W) and phenylalanine (F), which form the core of this binding groove. Rab27A and Rab31 are particularly critical for MVE docking and ILV formation. The knockout of Rab27A in colorectal cancer models significantly reduced exosome secretion and suppressed tumor growth [6]. Similarly, Yang Y. et al. indicated that inhibition of exosome secretion by Rab27a knockdown in tumor cells inhibited the growth of 4T1 mouse mammary tumor cells, similar to GW4869 treatment [9].

Interestingly, combining Rab27A knockout with anti-PD-L1 therapy produced additive effects, highlighting the complementary roles of ExoPD-L1 inhibition and immune checkpoint blockade [6]. However, the systemic inhibition of exosome secretion raises concerns about off-target effects on normal cellular communication. Future research should prioritize tumor-specific delivery methods for these inhibitors.

### 5.2. Neutralizing Exosomal PD-L1 with Antibodies

Neutralizing antibodies designed to bind specifically to ExoPD-L1 offers a direct approach for blocking its immunosuppressive function. However, a major challenge lies in ensuring antibody specificity for cancer-specific exosomes to prevent potential side effects. Engineering dual-targeting antibodies that recognize both PD-L1 and exosome-specific markers, such as CD63, CD9, and CD81, could enhance precision while minimizing systemic toxicity [70].

Another approach involves bispecific antibodies classified as T cell engagers that prevent the interaction between ExoPD-L1 and PD-1 on T cells, preserving T cell activation and cytokine production. For example, studies have shown that neutralizing bispecific antibodies targeting both PD-L1 and CD 3 (cluster of differentiation 3) can reverse T cell suppression mediated by tumor-derived exosomes in vivo, enhancing their migration and anti-tumor activity. In murine B16 melanoma, MC38 colon cancer, and human multiple myeloma cells, these bispecific antibodies significantly migrated into tumors, attracted by the presence of PD-L1 in tumor cells and by the secretion of Exo-PD-L1 [68]. Consequently, an increase in infiltrating effector memory CD8+ T cells into tumors was noticed, confirming the potential of bispecific antibodies in tumor therapy.

Currently, there are numerous clinical trials involving immune-modulating bispecific antibodies classified as T cell engagers in phase I/II clinical trials for treating hematological and solid tumors [71]. However, neurotoxicity and cytokine release syndrome seem to be two major adverse events following bispecific antibody therapy [71,72].

### 5.3. Silencing PD-L1 Expression via RNAi and CRISPR

Two emerging gene-silencing technologies, RNA interference (RNAi) and CRISPR/Cas9 gene editing, have shown promise in reducing PD-L1 expression at its source, enhancing immunotherapy outcomes.

RNAi-mediated knockdown of PD-L1 mRNA in tumor cells has demonstrated efficient post-transcriptional silencing of PD-L1 in various tumor models. For example, Teo PY et al. showed that siRNA-mediated PD-L1 knockdown in ovarian cancer cells sensitizes them to T cells [73]. Similarly, Ligtenberg et al. applied RNAi to downregulate PD-1 in T cells, indirectly reducing PD-L1 signaling, which led to enhanced adoptive T cell therapy responses in melanoma and increased T cell capacity to secrete IFN-γ upon polyclonal stimulation [74].

CRISPR-based approaches extend this concept by enabling the permanent disruption of the PD-L1 gene. Abounar et al. and Fierro J. et al. successfully used CRISPR/Cas9 to knock out PD-L1 in lung and glioblastoma cells, respectively, leading to inhibition of tumor growth and invasion, with an enhanced immune response [75,76].

Interestingly, Deng et al. showed that targeting CDK5, an upstream regulator of PD-L1 transcription, via CRISPR also led to a significant reduction in PD-L1 levels in murine melanoma, lung metastasis, and in triple-negative breast cancer, indicating that both direct and indirect CRISPR strategies are viable [77]. X Yuan et al. suggested that hypoxia is another upstream regulator of ExoPD-L1 in metastatic nasopharyngeal carcinoma, via HIF-1α signaling pathway [30].

Wei et al. advanced the delivery of CRISPR cargo by designing nucleobase-modified polyamidoamine carriers, ensuring precise PD-L1 editing with minimal off-target effects [78]. Furthermore, Cheng et al. delivered both siRNA and CRISPR cargo using human serum albumin nanoparticles, showing that siRNA was effective at transient PD-L1 knockdown, while CRISPR offered a more permanent solution [79]. Notably, Gondaliya et al. compared siRNA and CRISPR strategies in cholangiocarcinoma models, reporting that siRNA provided faster onset but shorter-lasting suppression of PD-L1 [80].

However, challenges remain regarding the safe delivery of RNAi or CRISPR constructs in vivo. The main limitation of gene silencing is delivery. Since the oligonucleotides are negatively charged, their internalization is difficult, and the RNAs are vulnerable to RNases in circulation, so they will be rapidly degraded. Another method, based on viral vector transfection, although it has high efficiency, it raises biosafety concerns. It remains only methods that use carriers such as calcium carbonate/phosphate, inert gold nanoparticles, carbon/silicon-based nanomaterials, hydroxide nanoparticles, various polymers, and positively charged lipids [81,82,83]. The majority of these gene silencing drugs are currently tested in vitro and in vivo, being optimized for efficient tumor-specific delivery.

In conclusion, blocking ExoPD-L1 represents a novel approach to overcoming immune evasion mechanisms in cancer. Strategies such as inhibiting exosome biogenesis with GW4869 or Rab inhibitors, neutralizing ExoPD-L1 with targeted antibodies, and silencing PD-L1 expression via RNAi or CRISPR hold significant therapeutic potential. While each approach has its limitations, their integration into combination regimens offers a promising path forward for enhancing the efficacy of immune checkpoint therapies. Future research should prioritize optimizing these strategies for clinical use while addressing challenges related to delivery specificity and off-target effects.

## 6. Challenges and Future Directions

Exosomes reflect the physiological and pathological state of the cells of origin through the molecules and biomarkers they carry, and PD-L1, a molecule involved in the suppression of the immune response and the immune evasion of tumor cells, is one of them.

We already know that tumors are not homogeneous, and their heterogeneity underlies complexity. This intra-tumoral heterogeneity is associated with an ExoPD-L1 profile depending on the sampling site, disease stage, or even the time of day, challenging the reliability of single-point measurements. On the other hand, many factors, including tumor type, genetic background, immune microenvironment, and therapeutic history, shape inter-patient variability. Even among patients with the same cancer subtype, ExoPD-L1 levels can differ markedly, complicating the interpretation of its prognostic or predictive value. Taking into account the inter- and intra-tumor variability of PD-L1, as well as the limitations in terms of sensitivity/specificity (PD-L1 can also be expressed in other inflammatory conditions), the heterogeneity of exosomes due to their origin from both tumor cells and immune system cells or other tissues, and the ability of tumors to use alternative pathways to avoid the immune system (tumor immune evasion), the need for a multi-marker approach is essential for more accurately assessing immune suppression and developing a more robust and predictive biomarker profile. Therefore, a series of complementary markers may be helpful alongside ExoPD-L1, such as: exosomal CD63/CD9/CD81, which confirm exosomal purity; ExoPD-L1 that reflects T cell exhaustion or a suppressed immune status; immune checkpoints (ICIs) such as exosomal CTLA-4, TIM-3, and LAG-3, which indicate tumor immune evasion and adaptive resistance; other markers for inflammation, proliferation, and tumor angiogenesis, such as miRNAs, EGFR, HER2, KRAS mutations, lack of MHC I/MHC II; and the cytokine profiles that can highlight a TME with an inflammatory status often involved in resistance to ICIs. Integrating these markers through multi-omic profiling and machine learning provides a more comprehensive and dynamic view of tumor-immune interactions, enhancing the predictive value of exosomal biomarkers in cancer treatment.

Alongside the potential of exosomal biomarkers to refine cancer diagnostics and treatment monitoring, attention is now turning toward their therapeutic targeting, particularly in the context of PD-L1-mediated immune suppression. Given that PD-L1-enriched tumor-derived exosomes suppress T cell activity systemically, thus contributing to immune evasion even in patients undergoing anti-PD-1/PD-L1 therapy, targeting ExoPD-L1, either by inhibiting its biogenesis, release, or function, could restore immune surveillance and enhance the efficacy of existing immunotherapies. Technically, the approach is complex, requiring precise strategies to distinguish between exosomal and membrane-bound PD-L1 and robust detection methods to monitor therapeutic response. Moreover, the combination of different therapies with exosome-based strategies could be a strategic approach to overcome resistance mechanisms in cancer treatment and improve clinical outcomes.

Given the biological complexity of combination therapies, several obstacles need to be overcome, including drug loading efficiency, standardization of isolation and characterization methods, safe delivery, and policy regulation of these strategies. An interdisciplinary collaboration between immunologists, bioengineers, and clinicians, as well as advances in nanotechnology and rigorous validation of exosome-targeting therapies, are needed to successfully implement this transformative step towards more effective and individualized cancer treatment.

## Figures and Tables

**Figure 1 cimb-47-00525-f001:**
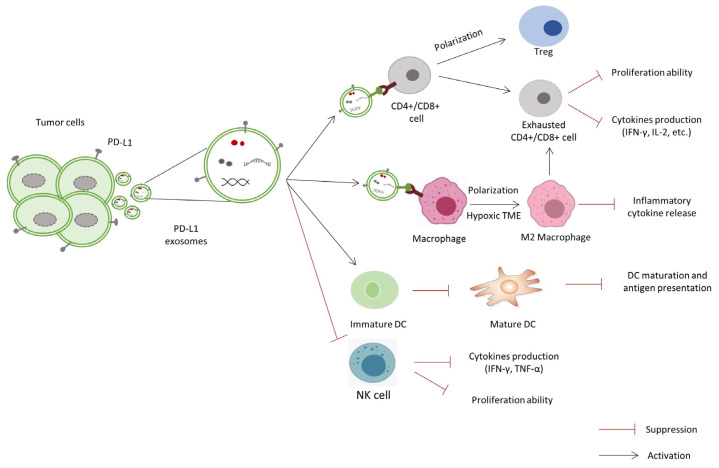
A schematic representation of tumor-derived ExoPD-L1 and immune cells interactions. ExoPD-L1 inhibits CD8+ T cell proliferation, cytokine production, and promotes the proliferation and function of Tregs. The specific effects of tumor-derived ExoPD-L1 on other immune cells: the polarization of M2 macrophages, the inhibition of DCs differentiation and maturation, and the suppression of NK cells’ immune ability.

**Figure 2 cimb-47-00525-f002:**
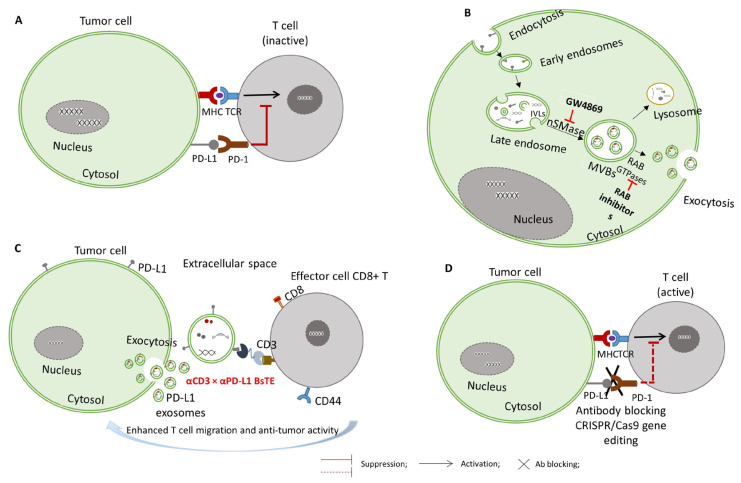
Therapeutic strategies for targeting the PD-1/PD-L1 axis: (**A**) the PD-1/PD-L1 interaction: PD-L1+ tumor cells or ExoPD-L1 interact with T cells by binding to PD-1 receptors, leading to T cell inhibition and reduced activation, thus promoting immune evasion. (**B**) Inhibiting the extracellular release of ExoPD-L1 through the administration of small molecule inhibitors (e.g., GW4869, RabGTPase inhibitors). GW4869 targets nSMase2, disrupting ceramide-dependent ILV formation within MVBs and reducing the exosome release. RabGTPase inhibitors target key regulators of exosome trafficking, being involved in the fusion of the MVB with the plasma membrane. (**C**) Anti-tumor activity of αCD3 × αPD-L1 bispecific antibody-armed T cells–ExoPD-L1 and PD-L1 expression in tumors play critical roles in BsTE:T migration and anti-tumor activity. Bystander CD8+ T cells also contribute to the elimination of tumor cells. (**D**) Targeting PD-1/PD-L1 expression by CRISPR/Cas9 genome editing of PD-L1 or by using antibodies.

**Table 1 cimb-47-00525-t001:** Structure of several exosome inhibitors and bispecific antibodies targeting ExoPD-L1.

Compound/Antibody	Target	Mechanism/Recent Application (2020–2025)	Structure ^#^	References
GW4869	nSMase2/exosomes	Inhibits nSMase2; reduces ceramide biosynthesis, leading to decreased exosome release; widely used in cancer/inflammation models.	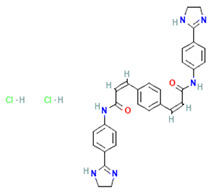	[62,63]
Nexinhib20	Rab27A	Blocks Rab27A–JFC1 interaction; inhibits EV release in cancer and immune cells; pharmacologically validated.	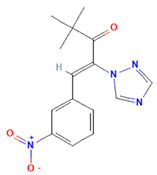	[63,64]
Tipifarnib	Rab27A (modeling)	Used in molecular docking simulations targeting Rab27A ‘WF-pocket’; proposed structure-based inhibitor.	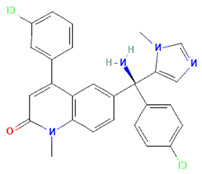	[65,66]
Y-27632	ROCK1/2 (indirect exosomes)	Inhibits ROCK kinase; decreases EV/microvesicle release in models of cancer and inflammation.	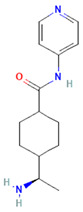	[67]
BsAb anti-PD-L1/CD63	PD-L1 + exosomal marker CD63	Bispecific DVD-Ig; simultaneously binds PD-L1 (immune checkpoint) and CD63 (exosome marker); enables selective targeting of exo-PD-L1.	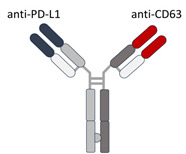	[68]

^#^ The structure of GW4869, Nexinhib20, Tipifarnib, and Y-27632 were obtained from the National Center for Biotechnology Information (2025)—PubChem [69].

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
