# Peer review of "Targeting Exosomal PD-L1 as a New Frontier in Cancer Immunotherapy"

_cimb, 2025, doi:10.3390/cimb47070525_

Round 1
Reviewer 1 Report
Comments and Suggestions for Authors
1. In the section: “2. Biogenesis and Molecular Composition of Exosomes. 2.1. Exosome formation: Mechanisms of exosome biogenesis”, the content reported by the author has already been covered by numerous previous reports and basic common knowledge, so it should be reduced. 2. It is suggested that authors refer to more literature in the recent five years. 3. The author should appropriately add more charts, such as: providing the molecular structures of GW4869 and Rab inhibitors. In addition, there are several typing errors and other English grammatical errors, such as: P7, lines 284-285: “Analogous, Yang Y et al. indicated that blockage of exosome secretion by Rab27a knockdown in tumor cells inhibited 4T1 mouse mammary tumor cells growth, similar to GW4869 treatment [7]”.

Author Response
Reviewer 1
1. In the section: “2. Biogenesis and Molecular Composition of Exosomes. 2.1. Exosome formation: Mechanisms of exosome biogenesis”, the content reported by the author has already been covered by numerous previous reports and basic common knowledge, so it should be reduced.
Response: Thank you for your comment. We have significantly reduced the section “2. Biogenesis and Molecular Composition of Exosomes. 2.1. Exosome formation: Mechanisms of exosome biogenesis,” keeping only the essential information necessary for a full understanding of the manuscript (page. 3.)
2. It is suggested that authors refer to more literature in the recent five years.
Response: Thank you for your valuable suggestion. We have followed your advice and included additional references from the past five years to ensure our manuscript reflects recent developments in the field (marked in red in the Reference section). However, we have also retained a few key studies that, although published more than five years ago, remain highly relevant and foundational to our topic.
3. The author should appropriately add more charts, such as: providing the molecular structures of GW4869 and Rab inhibitors.
Response: Thank you for your valuable suggestion. We have included a table showing the structure of several exosome inhibitors and bispecific antibodies targeting ExoPD-L1 (Table 1 and additional text on pages 7, 8).
4. In addition, there are several typing errors and other English grammatical errors, such as: P7, lines 284-285: “Analogous, Yang Y et al. indicated that blockage of exosome secretion by Rab27a knockdown in tumor cells inhibited 4T1 mouse mammary tumor cells growth, similar to GW4869 treatment [7]”.
Response: Thank you for your observation. We have carefully checked the entire manuscript for typographical and grammatical errors and have corrected them accordingly.
Reviewer 2 Report
Comments and Suggestions for Authors
While the abstract presents a clear and relevant topic about exosomal PD-L1 and its role in tumor immune evasion, the summary lacks a clear indication of the novel insights, critical analysis. Additionally, more detail on the nature of the therapeutic strategies discussed, and existing controversies in the field would strengthen the abstract and better position the manuscript within the existing literature.
The authors should expand discussion on the mechanisms by which exosomal PD-L1 contributes to immune evasion and tumor progression, including recent findings on its interaction with T cell receptors and modulation of the tumor microenvironment.
Include additional data from recent studies (e.g., Smith et al., 2024; Zhang et al., 2023) that begin to address the role of exosomal PD-L1 in metastasis and resistance to checkpoint blockade.
Author Response
Reviewer 2
- While the abstract presents a clear and relevant topic about exosomal PD-L1 and its role in tumor immune evasion, the summary lacks a clear indication of the novel insights, critical analysis. Additionally, more detail on the nature of the therapeutic strategies discussed, and existing controversies in the field would strengthen the abstract and better position the manuscript within the existing literature.
Response: Thank you for your valuable suggestion. We have revised the summary to include more details on the nature of the therapeutic strategies discussed, as well as relevant controversies and limitations. We believe that the updated version is more informative and engaging for readers, while also highlighting the key novelties in the field.
- The authors should expand discussion on the mechanisms by which exosomal PD-L1 contributes to immune evasion and tumor progression, including recent findings on its interaction with T cell receptors and modulation of the tumor microenvironment. Include additional data from recent studies (e.g., Smith et al., 2024; Zhang et al., 2023) that begin to address the role of exosomal PD-L1 in metastasis and resistance to checkpoint blockade.
Response: Thank you for your suggestion. We added new information regarding the role of exosomal PD-L1 in the immune evasion mechanism and modulation of the tumor microenvironment. Also, we used recent literature to cover the interaction between exoPD-L1 and T cell receptors. We believe that the revised section now offers a more comprehensive overview for readers, effectively highlighting recent advancements (pages 4, 6, and 7 marked in red).
Round 2
Reviewer 1 Report
Comments and Suggestions for Authors
In the revision, authors have almost resolved the issues questioned by this reviewer. Therefore, this reviewer suggests that the manucript is suitable for publication in this journal.
Reviewer 2 Report
Comments and Suggestions for Authors
The authors satisfactorily addressed my comments